# Metabolite Variation between Nematode and Bacterial Seed Galls in Comparison to Healthy Seeds of Ryegrass Using Direct Immersion Solid-Phase Microextraction (DI-SPME) Coupled with GC-MS

**DOI:** 10.3390/molecules28020828

**Published:** 2023-01-13

**Authors:** Pushpendra Koli, Manjree Agarwal, David Kessell, Shalini Mahawar, Xin Du, Yonglin Ren, Simon J. McKirdy

**Affiliations:** 1Harry Butler Institute, Murdoch University, Murdoch, WA 6150, Australia; 2Indian Council of Agricultural Research (ICAR)-Indian Grassland and Fodder Research Institute, Jhansi 284003, India; 3College of Science, Health, Engineering and Education, Murdoch University, Murdoch, WA 6150, Australia; 4Scientific Service Division, ChemCentre, Western Australia, Bentley, WA 6102, Australia; 5Department of Primary Industries and Regional Development, Perth, WA 6151, Australia

**Keywords:** annual ryegrass, ARGT, bacteria, DI-SPME, galls, GC-MS, metabolites, nematode, toxicity ryegrass

## Abstract

Annual ryegrass toxicity (ARGT) is an often-fatal poisoning of livestock that consume annual ryegrass infected by the bacterium *Rathayibacter toxicus*. This bacterium is carried into the ryegrass by a nematode, *Anguina funesta*, and produces toxins within seed galls that develop during the flowering to seed maturity stages of the plant. The actual mechanism of biochemical transformation of healthy seeds to nematode and bacterial gall-infected seeds remains unclear and no clear-cut information is available on what type of volatile organic compounds accumulate in the respective galls. Therefore, to fill this research gap, the present study was designed to analyze the chemical differences among nematode galls (*A. funesta*), bacterial galls (*R. toxicus*) and healthy seeds of annual ryegrass (*Lolium rigidum*) by using direct immersion solid-phase microextraction (DI-SPME) coupled with gas chromatography–mass spectrometry (GC-MS). The method was optimized and validated by testing its linearity, sensitivity, and reproducibility. Fifty-seven compounds were identified from all three sources (nematode galls, bacterial galls and healthy seed), and 48 compounds were found to be present at significantly different (*p* < 0.05) levels in the three groups. Five volatile organic compounds (hexanedioic acid, bis(2-ethylhexyl) ester), (carbonic acid, but-2-yn-1-yl eicosyl ester), (fumaric acid, 2-ethylhexyl tridec-2-yn-1-yl ester), (oct-3-enoylamide, N-methyl-N-undecyl) and hexacosanoic acid are the most frequent indicators of *R. toxicus* bacterial infection in ryegrass, whereas the presence of 15-methylnonacosane, 13-methylheptacosane, ethyl hexacosyl ether, heptacosyl acetate and heptacosyl trifluoroacetate indicates *A. funesta* nematode infestation. Metabolites occurring in both bacterial and nematode galls included batilol (stearyl monoglyceride) and 9-octadecenoic acid (Z)-, tetradecyl ester. Among the chemical functional group, esters, fatty acids, and alcohols together contributed more than 70% in healthy seed, whereas this contribution was 61% and 58% in nematode and bacterial galls, respectively. This study demonstrated that DI-SPME is a valid technique to study differentially expressed metabolites in infected and healthy ryegrass seed and may help provide better understanding of the biochemical interactions between plant and pathogen to aid in management of ARGT.

## 1. Introduction

Ryegrass is an important pasture and forage plant that is widely cultivated around the world to feed grazing livestock [1,2]. It is highly palatable, protein-rich, and has high productivity and excellent nutritional value [3]. Two types of ryegrasses are routinely cultivated for this purpose: annual ryegrass and perennial ryegrass. Perennial ryegrass (*Lolium perenne*) is densely tufted, multi-tillered and grows fastest in autumn and spring, whereas annual ryegrass species (*Lolium rigidum, L. multiflorum*) grow in the winter–spring period [4]. In future, annual ryegrass may replace fodder oat and other forage crops, because of their longer growing period and improved feed quality [5]. A constraint on annual ryegrass adoption is annual ryegrass toxicity (ARGT), present in some regions of South Australia and Western Australia [6]. This fatal disease is caused by association of the nematode *Anguina funesta* and the bacteria *Rathayibacter toxicus* (formerly known as *Clavibacter toxicus*) [7,8]. Plant infestation may result in the formation of two types of galls—one by the nematode and the other by the bacteria—during the period between flowering and seed maturity. The toxin is produced in bacterial galls. Ryegrass containing bacterial galls remains toxic even when senescent and dried off, so hay made from infected ryegrass is also toxic. All grazing animals are susceptible, including horses and pigs [9]. The effects of toxic galls on livestock have been reported in South Africa [10] and Japan [11], and seed galls obtained from New Zealand were positive for the presence of the *R. toxicus* antigen [12], but it is most serious in Australia, where it has been responsible for the deaths of hundreds of thousands of sheep and cattle [6,13]. In Western Australia, a 2004 report highlighted annual wool production was reduced by 3% because of ARGT [6].

The association of the nematode and bacterium that causes ARGT is widely studied [7,14,15,16,17]. A bacterial inhibition assay was developed to detect toxicity [18,19]. High-performance liquid chromatography (HPLC)-based analytical methods detected corynetoxins [20]. An enzyme-linked immunosorbent assay (ELISA) is widely used to detect water-soluble antigens specific to *R. toxicus* [21]. Polymerase chain reaction (PCR) in combination with corynetoxin ELISA and real-time PCR is used to detect the presence of *R. toxicus* and its associated phage [22,23,24]. These assays are used to predict disease occurrence. However, no one method meets all the regulatory requirements [25]. All the above methods focus on the bacteria and none on the nematode, even though the nematode plays an important role in carrying the bacterium into the plant. Therefore, to understand the chemical interaction between nematode and bacteria and the ryegrass plant, information on chemicals present is required from both the bacterial and nematode galls. In this study, direct immersion solid-phase microfiber extraction (DI-SPME) coupled with gas chromatography (GC)–mass spectrometry (MS) (GC-MS) was used to study the chemical profile of bacterial and nematode galls. The advantages of this method include rapid sample preparation, and it is an efficient method to detect and separate analytes. DI-SPME is usually coupled with GC, GC–MS, HPLC or LC–MS for accurate analysis. DI-SPME has advantages over other methods of low solvent use and disposal costs, and improved limits of detection. However, one of the drawbacks of DI-SPME is associated with its handling, such as bending, breaking and stripping of coating that can decrease its life span and that can be overcome by using HS-SPME to a certain limit [26]. DI-SPME has been used in a wide range of applications, including identification of biomarkers, food safety and quality control, food analysis, the pharmaceutical industry, process monitoring, and forensics [27,28,29,30].

This study applied DI-SPME coupled with GC-MS to gain information on metabolic profiles associated with nematode and bacterial galls in annual ryegrass to better understand the biochemical interactions between the nematode, bacterium, and plant. This research aims to provide information for the development of detection assays and improve management of disease.

## 2. Results

### 2.1. Identification of Metabolites

Fifty-seven compounds were detected from the three treatment sources: 30 from nematode galls, 25 from bacterial galls and 24 from healthy seed (Table 1). All the annual ryegrass material was collected from a farmer’s field at Bindoon, Western Australia in 2020, and nematode and bacterial galls were separated for analysis. Forty-eight compounds were found to be significantly different (*p* < 0.05) upon one-way ANOVA and post hoc analysis (Fisher’s LSD) (Figure 1). Some compounds were common to all three treatment sources, while others were differentially expressed in one or two treatment sources (Figure 2). The compounds bicyclo[2.2.1]heptane-2-carboxylic acid, 3,3-dimethyl-, methyl ester; 7-methyl-1-naphthol; 2,4-di-tert-butylphenol; 4-(2-(acryloyloxy)ethoxy)-4-oxobutanoic acid, TMS; carbonic acid, eicosyl vinyl ester; 4′,6′-dimethoxy-2′-hydroxychalcone, and 2-methylpropionate were common to all three treatments, with two compounds, batilol and 9-octadecenoic acid (Z)-, tetradecyl ester, expressed in both the nematode and the bacterial galls. Seven compounds—2-cyclohexen-1-ol; (1R)-1-(2,6-dichloro-3-fluorophenyl) ethanol, methyl ether; n-hexadecanoic acid; pent-4-enoyl amide, 2-methyl-N-dodecyl-; nonadecane, 1-chloro-; 2,4-dihydroxyheptadecyl acetate; and isopropyl hexacosyl ether—were expressed in healthy seed and bacterial galls, but not in nematode galls, and one compound (*cis*-10-nonadecenoic acid) was expressed in nematode galls and healthy seed, but not in bacterial galls. There were 21, 10 and 10 unique compounds expressed in nematode galls, bacterial galls, and healthy seed, respectively. In nematode galls, some important compounds were found in large quantities compared to others, suggesting a direct connection to the presence of nematodes, i.e., octadecenoic acid; nonadecanoic acid; octadecynoic acid, methyl ester; carbonic acid, octadecyl vinyl ester; fumaric acid, pent-4-en-2-yl tridecyl ester; ethyl hexacosyl ether; 15-methylnonacosane; 13-methylheptacosane; heptacosyl acetate; dotriacontane, 2-methyl-; propyl tetracosyl ether; 1-docosanol, acetate and fumaric acid, hexadecyl 4-heptyl ester were found at significant levels. The compounds hexanedioic acid, bis(2-ethylhexyl) ester; carbonic acid, but-2-yn-1-yl eicosyl ester; hexacosanoic acid; oct-3-enoylamide, N-methyl-N-undecyl-; estra-1,3,5(10)-trien-17β-ol and tetradecanoic acid were present only in bacterial galls. In healthy seed, the most statistically significant (*p* < 0.05) compounds were hexanedioic acid, dioctyl ester; pent-4-enoyl amide, 2-methyl-*N*-tetradecyl-; 9-hexadecen-1-ol, (Z)- 2-phosphonopropanoic acid, 3TMS; hexadecyl acrylate. The representative chromatogram and mass spectra of each group are presented in the Appendix A.

### 2.2. Multivariate Analysis to Classify Nematode Galls, Bacterial Galls and Healthy Seeds

Distribution of compounds in the three treatment sources was analyzed using principal component analysis (PCA) (Figure 3) and a hierarchical cluster analysis heat map (Figure 4). PCA is an unsupervised method that helps to visualize covariance and correlation among the three treatment sources and metabolites expressed in each.

There is clustering of metabolites from the three replicates of each treatment source, and the three treatment ellipses do not overlap (Figure 3). This separation indicates that the three treatment sources have different chemical profiling, and this profiling can be used as a marker to detect nematode and bacterial infection in annual ryegrass. The first principal component explains 62.2% of the variance, whereas the second principal component explains 33.2%. Similarly, Figure 4 shows the cluster and heat map of all three treatment sources used in this study. The heat map shows a clear expression level difference between healthy seed, nematode and bacterial galls.

Further, to know the differential compounds between the individual groups, supervised analysis was conducted. Partial least squares-discriminant analysis (PLS-DA) and orthogonal PLS-DA (OPLS-DA) score plots were drawn between nematode gall versus healthy seed and bacterial gall versus healthy seed (Figure 5). The compounds 2-cyclohexen-1-ol and 1-hexadecanol, 2-methyl clearly distinguish healthy seeds from nematode galls by showing their abundance in healthy seeds, whereas due to the abundant nature of carbonic acid, but-2-yn-1-yl eicosyl ester and fumaric acid, 2-ethylhexyl tridec-2-yn-1-yl ester in the bacterial gall, they were separated from the healthy seeds.

Chemical functional groups differed among the three. Esters, fatty acids, and alcohols were the dominant groups in healthy seed, representing 34%, 17%, and 13% of the total number of compounds, respectively, followed by amides, ethers, hydrocarbons, and phenols, each at 8%, and heterocyclic compounds at 4% (Figure 6). In nematode galls, the proportions of esters, hydrocarbons, and acids were 48%, 20%, and 10%, respectively, followed by ethers and phenols, each at 7%, and alcohols and aldehydes, each at 3%. In bacterial galls, the proportions of esters, acids, alcohols, hydrocarbons were 36%, 16%, 12%, and 12%, respectively, followed by amides, ethers, and phenols, with 8% in each group. Interestingly, no amides and only the aldehydes were present in the nematode galls, and heterocyclic compounds were found only in healthy seed. In addition to this, esters were highly expressed in each treatment, and the fraction of hydrocarbons was greater in infected seed than healthy seed.

## 3. Discussion

To optimize the DI-SPME extraction protocol, the fiber was exposed at room temperature (25 ± 2 °C) for the analyte extraction for different times (30, 45, 60 and 120 min) and desorption time (3 min, 6 min and 10 min). The optimum extraction time, 60 min, and desorption time, 6 min, were optimized based on the number of peaks and peak areas. Similarly conditioned optimization parameters for SPME extraction were reported for VOC detection in barley [31].

Nontargeted metabolite analysis revealed relative expression levels of compounds present in the sample. This enables identification of the prominent compounds in each treatment, information that can be used to develop diagnostic assays and inform breeding efforts. Bacterial gall samples expressed 2,4-dihydroxyheptadecyl acetate derivatives, and this has been described previously [32,33]. Fatty acids play an important role in bacterial lipid membranes, and it has previously been reported that the presence of C-15 saturated branched chain acid, pentadecanoic acid, C-17 saturated branched chain acids and normal saturated fatty acids, such as lauric, myristic, stearic and arachidic, are major components of cell membranes in *Rathayibacter* species. [34,35,36]. Here, the closely related fatty acids tetradecanoic acid (myristic), hexadecenoic acid, and hexacosanoic acid were identified in bacterial galls. The presence of oct-3-enoyl amide, *N*-methyl-*N*-undecyl-, a structural derivative of an alpha-amino acid, and may be responsible in biological pathways for producing 2,4-diaminobutyric acid, which is a characteristic of *R. toxicus* [37]. Maulidia et al. [38] reported steroidal compounds as the major component in methanolic extracts of Gram-positive *Bacillus thuringiensis* associated with root knot nematodes, and similarly, this study shows the presence of stigmastane-3,6-dione (5α), which is classified as a steroid compound, in Gram-positive *R. toxicus* bacterial galls. Fumaric acid is an important intermediate of the tricarboxylic acid (TCA) cycle in plant systems [39]. Expression of fumaric acid, 2-ethylhexyl tridec-2-yn-1-yl ester, in the bacterial galls suggests that this bacterium could be interfering with the host TCA cycle.

There is no published information available on volatile organic compounds generated in nematode-infested cereal seed galls. However, there are reports that showed the protein content is reduced during nematode infestation in ear cockle disease of wheat [40]. Similarly, compounds containing amide groups, such as pent-4-enoyl amide, 2-methyl-*N*-dodecyl-, pent-4-enoyl amide, 2-methyl-*N*-tetradecyl- and oct-3-enoylamide, and *N*-methyl-*N*-undecyl- were downregulated in nematode gall. Compounds expressed by soybean cyst nematode and analyzed with GC-MS were methyl ester derivatives, a component of nematode pheromones [41]. In our study, the presence of the derivatives of methyl, ethyl and vinyl esters in the nematode galls could indicate the presence of pheromones associated with *A. funesta*. Nematodes are also known to contain significant quantities of lipids in their body cuticle [42], and fatty acids and fatty aldehydes are major components of the nematode cuticle [43]. Phospholipids play an important role in the formation of nematode cellular membranes and in signal transduction pathways [44]. In our results, the presence of octadecanal, which is a phospholipid derivative [45], could be a potential marker to identify *A. funesta*. Wei et al. [46] mentioned the role of 17-octadecynoic acid during synthesis of phospholipids in *Caenorhabditis elegans,* suggesting that 17-octadecynoic acid, methyl ester and (*E*)-9-octadecenoic acid ethyl ester may play a role in the development of the nematode in the nematode gall. Among the fatty acids, the presence of saturated fatty palmitic (16:0) and stearic (18:0) acids were reported from the parasitic nematode *Ascaridia galli* [43], whereas we found the unsaturated fatty octadecenoic acid (18:1) and nonadecanoic acid (19:1), which may indicate different requirements for fatty acids in *A. funesta*.

The high concentration of phenolic compounds in healthy seeds is noteworthy. Hexanedioic acid, dioctyl ester and 9-hexadecen-1-ol, (Z) are considered as bioactive phytochemicals and are known nematicides [47,48].

## 4. Materials and Methods

### 4.1. Collection of Infected Plant Materials

Infected annual ryegrass samples were collected in December 2020 from a farmer’s field at Bindoon, Western Australia (31°21′2″ S 116°7′47″ E). The ryegrass seeds along with galls were separated using sieves (1.4 mm, 1.70 mm and 850 µm) by manual threshing in the laboratory. After separation, the bacterial galls, nematode galls and healthy seeds were physically identified and differentiated using a light box [9]. All three were presorted based on their shape and color under the light box (Figure 7). The nematode gall appears dark brown to blackish with a pointed end, bacterial gall looks yellow in color with a piercing end, whereas healthy seed is denser and exhibits normal curvature, as shown in Figure 7.

### 4.2. Sample Preparation and Extraction Using DI-SPME

Twenty-five milligrams of each treatment source of plant sample (10–15 seeds or galls without removal of lamella or the gall walls) was transferred into 2 mL microtube containing 1 mL of HPLC grade acetonitrile. Three separate samples were prepared for each source. The microtube was sealed with a screw cap and shaken in a bedbug homogenizer (Benchmark Scientific, Shanghai, China), using two milling balls for 2 min at 4000 RCF, then another 1mL of acetonitrile was added, shaken for 2 min, and then centrifuged at 4000 RCF for 3 min using a minicentrifuge (Dynamica Velocity 13µ, Techcomp Europe, Livingston, UK). After that, the supernatant (~1.5 mL) was collected in a 2 mL GC vial with septum and stored at 4 °C until further use.

For conducting DI-SPME, a three-phase fiber (50/30 µm DVB/CAR/PDMS, Stableflex 2 cm, Supelco®, Bellefonte, PA, USA) was used. Three replicate sets of each sample were exposed to three different fibers. All new fibers were preconditioned for 60 min at 270 °C as per the manufacturer’s instructions. The SPME fiber was directly immersed by inserting into the GC vial containing the sample for 60 min at room temperature (25 ± 2 °C), and thereafter the fiber was taken out and injected directly into the GC-MS instrument with desorption for 6 min at inlet temperature 290 °C to analyze the sample.

### 4.3. GC-MS Conditions

The compounds were analyzed using a GC-MS 7890B coupled with a 5977B MSD mass spectrometer based on Agilent Technologies, Santa Clara, CA, USA with an Agilent HP-5MS column (30 m length, 0.25 mm internal diameter, 0.25 µm film thickness with 5% phenyl, and 95% dimethylpolysiloxane stationary phase). The following analytical conditions were used: splitless mode with helium as a carrier gas with a flow of 1 mL/min. The GC conditions were adopted from a previously used method [49], with slight modifications in temperature programming, starting at 50 °C with 5 min hold, then ramped at 6 °C/min to 90 °C, 8 °C/min to 140 °C, 6 °C/min to 190 °C, 4 °C/min to 240 °C, finally to 50 °C/min to 300 °C with hold at 300 °C for 12 min. The detector was operated in electron impact (EI) ionization mode at 70 eV and the spectra were acquired from 3 scan/s in a range from 50 to 550 atomic mass units (amu). The transfer line temperature of the MSD was 280 °C, and the ion source temperature was 230 °C with total run time of 51.95 min.

### 4.4. Data Processing and Statistical Analysis

For the identification of compounds, the National Institute of Standards and Technology Mass Spectrometry (NIST MS) was used with Kovat’s retention index (RI) of published data. The calculation for Kovat’s index is provided in the Appendix A. The data were processed with Agilent Mass Hunter Qualitative Analysis software (Version B.08.00, Build 8.0.8208.0, Santa Clara, CA, USA) [50]. The n-alkane (C_7_−C_40_, catalogue number 49451-U; Castle Hill, NSW, Australia) with concentration 100 µg/mL was used as the external standard. Microsoft Excel 2021 was used for data arrangement and other data processing. To perform multivariate analysis and principal component analysis (PCA) with one-way analysis of variance (ANOVA), MetaboAnalyst 5.0 (2022) (https://www.metaboanalyst.ca/MetaboAnalyst/upload/StatUploadView.xhtml; accessed on 20 June 2022) was used. R was used to draw the diagrams. Tukey’s post hoc test (HSD) was performed for statistical differences with *p* < 0.05).

## 5. Conclusions

Direct immersion solid-phase microextraction using a three-phase fiber was employed to collect volatile organic compounds from acetonitrile extract from nematode galls, bacterial galls, and healthy ryegrass seed, followed by analysis in GC-MS. The GC-MS analytical data showed a wide spectrum of metabolites in the three treatment sources. For the first time, this study has successfully identified chemical compounds that differ between nematode galls, bacterial galls, and healthy ryegrass seed. These three treatment sources could be distinguished based on their chemical signatures and could form the basis for development of new analytical tools for determining the risk of ARGT occurrence from feedstuff samples.

## Figures and Tables

**Figure 1 molecules-28-00828-f001:**
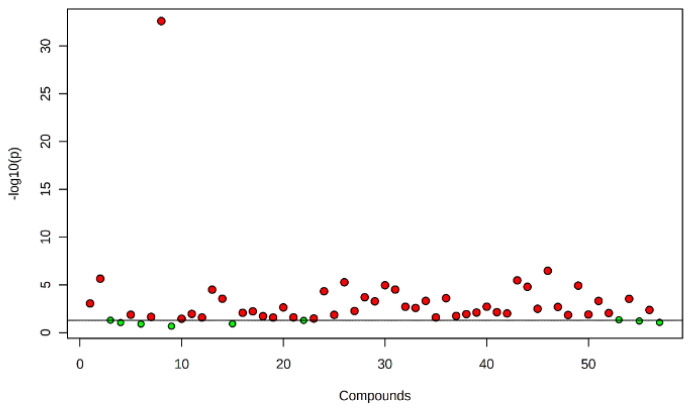
Metabolites obtained in nematode gall, bacterial gall and healthy seed of annual ryegrass. The points highlighted in red are significant compounds selected based on the *p*-value threshold (0.05), and green dots represent nonsignificant compounds.

**Figure 2 molecules-28-00828-f002:**
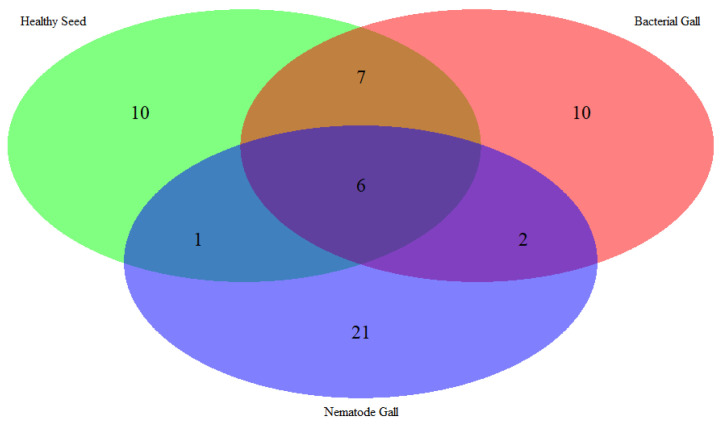
Venn diagram of the number of metabolites expressed in nematode gall (*Anguina funesta*), bacterial gall (*Rathayibacter toxicus*), and healthy seed of ryegrass (*Lolium rigidum*).

**Figure 3 molecules-28-00828-f003:**
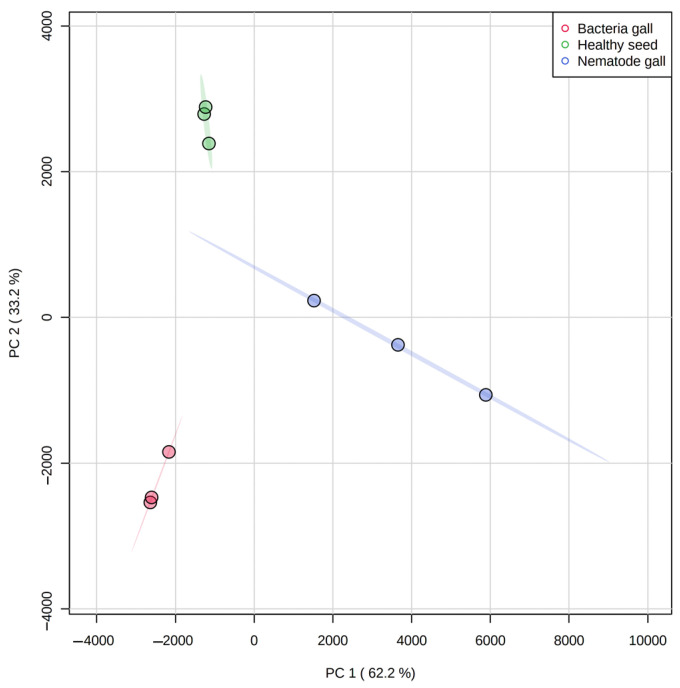
Principal component analysis (PCA) score plot for separation of metabolites in nematode gall (*Anguina funesta*), bacterial gall (*Rathayibacter toxicus*) and healthy seed of ryegrass (*Lolium rigidum*). The variances (PC1 and PC2) are shown in brackets. The three symbols for each treatment represent three biological replicates.

**Figure 4 molecules-28-00828-f004:**
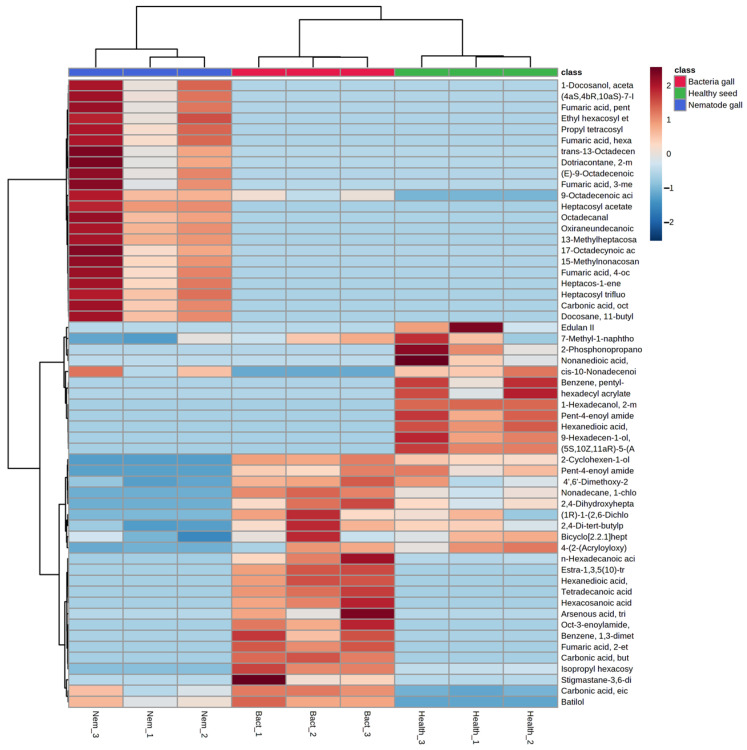
Heat map of metabolites obtained from nematode gall and bacteria gall in comparison to healthy seed of ryegrass. The three similarly colored columns represent biological replicates of each treatment. Red indicates Z-scores > 0 and blue indicates Z-scores < 0. The saturation threshold is set at ±2 (Z-score −2 to +2 representing low to high values).

**Figure 5 molecules-28-00828-f005:**
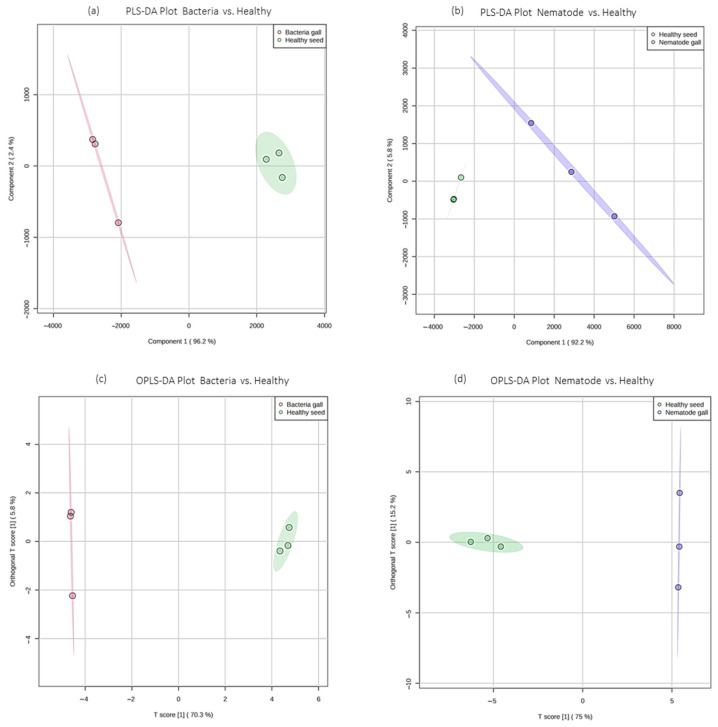
Partial least squares-discriminant analysis (PLS-DA) and orthogonal PLS-DA (OPLS-DA) score plots of ryegrass samples. (**a**,**b**) PLS-DA score plots for bacteria gall versus healthy seed and nematode gall versus healthy seed, respectively; (**c**,**d**) OPLS-DA score plots for bacteria gall versus healthy seed and nematode gall versus healthy seed, respectively. Red represents bacterial group, blue represents nematode group, and green represents healthy seeds for bacteria gall versus healthy seed and nematode gall versus healthy seed, respectively.

**Figure 6 molecules-28-00828-f006:**
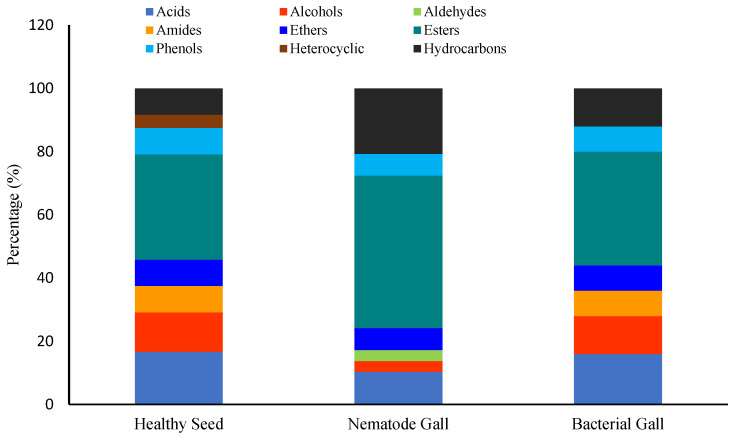
Ratios of chemical groups expressed in each treatment. Percentage is the ratio of the number of compounds of each group to the total number of compounds.

**Figure 7 molecules-28-00828-f007:**
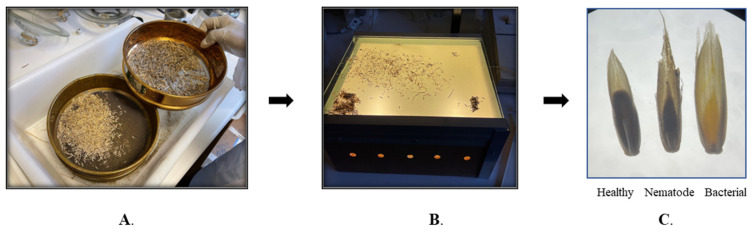
(**A**) Separation of heathy seeds and those with galls from ryegrass husk using different size of sieves, (**B**) light box used for identification and differentiation between infected galls and healthy seeds, (**C**) healthy seeds, nematode gall, and bacterial gall.

**Table 1 molecules-28-00828-t001:** Significant compounds’ peak areas (one unit corresponds to 10^4^ area) quantified in nematode galls, bacterial galls and healthy seeds of annual ryegrass seed using DI-SPME GC-MS.

Compounds	RI	RT	Treatment Sources			FDR	Molecular Formula	MW
			Nematode Gall	Bacterial Gall	Healthy Seed			
Benzene, 1,3-dimethyl-	876.8	8.09	N.D. ^b^	20.662 ^a^	N.D. ^b^	0.003	C8H10	106.08
2-Cyclohexen-1-ol	898.5	8.81	N.D. ^c^	10.620 ^a^	8.139 ^b^	0.000	C6H10O	98.07
Benzene, pentyl-	1161.7	16.27	N.D. ^b^	N.D. ^b^	20.270 ^a^	0.021	C11H16	148.13
(1R)-1-(2,6-Dichloro-3-fluorophenyl) ethanol, methyl ether	1296.6	19.05	N.D. ^b^	25.209 ^a^	13.113 ^ab^	0.031	C9H9Cl2FO	222.00
1-Hexadecanol, 2-methyl-	1315.9	19.43	N.D. ^b^	N.D. ^b^	13.948 ^a^	0.000	C17H36O	256.28
2-Phosphonopropanoic acid, 3TMS	1435.7	22.44	N.D. ^b^	N.D. ^b^	67.509 ^a^	0.042	C12H31O5PSi3	370.12
2,4-Di-tert-butylphenol	1519.6	23.26	27.074 ^b^	46.585 ^a^	39.439 ^a^	0.018	C14H22O	206.17
4-(2-(Acryloyloxy) ethoxy)-4-oxobutanoic acid, TMS	1648.9	25.77	48.480 ^b^	696.220 ^a^	856.414 ^a^	0.032	C12H20O6Si	288.10
Tetradecanoic acid	1758.0	27.84	N.D. ^b^	61.926 ^a^	N.D. ^b^	0.000	C14H28O2	228.21
9-Hexadecen-1-ol, (Z)-	1860.2	29.82	N.D. ^b^	N.D. ^b^	105.752 ^a^	0.001	C16H32O	240.25
n-Hexadecanoic acid	1966.7	31.97	N.D. ^b^	255.530 ^a^	27.134 ^b^	0.016	C16H32O2	256.24
Octadecanal	2021.2	33.08	43.277 ^a^	N.D. ^b^	N.D. ^b^	0.012	C18H36O	268.28
(4aS,4bR,10aS)-7-Isopropyl-1,1,4a-trimethyl-1,2,3,4,4a,4b,5,6,10,10a-decahydrophenanthrene	2067.6	34.03	74.366 ^a^	N.D. ^b^	N.D. ^b^	0.026	C20H32	272.25
hexadecyl acrylate	2098.4	34.66	N.D. ^b^	N.D. ^b^	16.190 ^a^	0.032	C19H36O2	296.27
Pent-4-enoyl amide, 2-methyl-N-dodecyl-	2141.7	35.53	N.D. ^b^	271.343 ^a^	271.581 ^a^	0.006	C18H35NO	281.27
17-Octadecynoic acid, methyl ester	2163.8	35.99	19.769 ^a^	N.D. ^b^	N.D. ^b^	0.032	C19H34O2	282.26
(E)-9-Octadecenoic acid ethyl ester	2175.0	36.22	101.540 ^a^	N.D. ^b^	N.D. ^b^	0.041	C20H38O2	310.29
Estra-1,3,5(10)-trien-17β-ol	2257.5	37.89	N.D. ^b^	68.765 ^a^	N.D. ^b^	0.000	C18H24O	256.18
cis-10-Nonadecenoic acid	2258.0	37.91	168.520 ^a^	N.D. ^b^	196.594 ^a^	0.021	C19H36O2	296.27
Nonadecane, 1-chloro-	2298.1	38.71	N.D.^c^	104.389 ^a^	51.739 ^b^	0.000	C19H39Cl	302.27
Carbonic acid, octadecyl vinyl ester	2299.3	38.74	93.518 ^a^	N.D. ^b^	N.D. ^b^	0.011	C21H40O3	340.30
Pent-4-enoyl amide, 2-methyl-N-tetradecyl-	2356.4	39.38	N.D. ^b^	N.D. ^b^	286.189 ^a^	0.001	C20H39NO	309.30
Oct-3-enoylamide, N-methyl-N-undecyl-	2356.4	39.38	N.D. ^b^	291.407 ^a^	N.D. ^b^	0.002	C20H39NO	309.30
Hexanedioic acid, dioctyl ester	2401.2	39.87	N.D. ^b^	N.D. ^b^	3569.522 ^a^	0.000	C22H42O4	370.31
Hexanedioic acid, bis(2-ethylhexyl) ester	2402.7	39.88	N.D. ^b^	4438.417 ^a^	N.D. ^b^	0.000	C22H42O4	370.31
Oxiraneundecanoic acid, 3-pentyl-, methyl ester, cis-	2433.4	40.09	67.235 ^a^	N.D. ^b^	N.D. ^b^	0.005	C19H36O3	312.27
2,4-Dihydroxyheptadecyl acetate	2466.5	40.31	N.D.^c^	78.065 ^a^	46.500 ^b^	0.006	C19H38O4	330.28
Carbonic acid, eicosyl vinyl ester	2499.6	40.54	744.725 ^b^	1370.199 ^a^	182.234^c^	0.002	C23H44O3	368.33
Fumaric acid, pent-4-en-2-yl tridecyl ester	2515.1	40.65	122.261 ^a^	N.D. ^b^	N.D. ^b^	0.032	C22H38O4	366.28
Batilol	2598.6	41.21	126.647 ^b^	187.007 ^a^	N.D.^c^	0.001	C21H44O3	344.33
1-Docosanol, acetate	2623.3	41.37	502.454 ^a^	N.D. ^b^	N.D. ^b^	0.025	C24H48O2	368.37
Fumaric acid, 4-octyl dodec-2-en-1-yl ester	2636.0	41.46	81.578 ^a^	N.D. ^b^	N.D. ^b^	0.019	C24H42O4	394.31
Heptacos-1-ene	2677.5	41.74	245.258 ^a^	N.D. ^b^	N.D. ^b^	0.015	C27H54	378.42
13-Methylheptacosane	2701.9	41.91	2675.940 ^a^	N.D. ^b^	N.D. ^b^	0.005	C28H58	394.45
4′,6′-Dimethoxy-2′-hydroxychalcone, 2-methylpropionate	2731.6	42.12	80.654 ^b^	232.646 ^a^	166.662 ^a^	0.014	C21H22O5	354.15
Propyl tetracosyl ether	2783.5	42.52	600.788 ^a^	N.D. ^b^	N.D. ^b^	0.017	C27H56O	396.43
Fumaric acid, 2-ethylhexyl tridec-2-yn-1-yl ester	2798.8	42.60	N.D. ^b^	328.787 ^a^	N.D. ^b^	0.000	C25H42O4	406.31
(5S,10Z,11aR)-5-(Acetyloxy)-6,10-bis(hydroxymethyl)-3-methylidene-2-oxo-2,3,3a,4,5,8,9,11a-octahydrocyclodeca[b]furan-4-yl 2-methylbutanoate, 2TMS derivative	2796.4	42.62	N.D. ^b^	N.D. ^b^	135.028 ^a^	0.000	C28H46O8Si2	566.27
Docosane, 11-butyl-	2799.5	42.64	340.807 ^a^	N.D. ^b^	N.D. ^b^	0.007	C26H54	366.42
Carbonic acid, but-2-yn-1-yl eicosyl ester	2819.3	42.80	N.D. ^b^	1032.923 ^a^	N.D. ^b^	0.000	C25H46O3	394.34
Heptacosyl trifluoroacetate	2879.9	43.31	708.013 ^a^	N.D. ^b^	N.D. ^b^	0.005	C29H55F3O2	492.42
Ethyl hexacosyl ether	2888.7	43.39	2961.463 ^a^	N.D. ^b^	N.D. ^b^	0.021	C28H58O	410.45
Isopropyl hexacosyl ether	2900.8	43.47	N.D.^c^	1546.260 ^a^	426.273 ^b^	0.000	C29H60O	424.46
15-Methylnonacosane	2902.0	43.50	3105.581 ^a^	N.D. ^b^	N.D. ^b^	0.020	C30H62	422.49
Hexacosanoic acid	2944.3	43.91	N.D. ^b^	213.320 ^a^	N.D. ^b^	0.002	C26H52O2	396.40
Fumaric acid, hexadecyl 4-heptyl ester	2986.6	44.33	223.079 ^a^	N.D. ^b^	N.D. ^b^	0.016	C27H50O4	438.37
Heptacosyl acetate	3081.7	45.39	1160.147 ^a^	N.D. ^b^	N.D. ^b^	0.001	C29H58O2	438.44
9-Octadecenoic acid (Z)-, tetradecyl ester	3478.0	48.29	256.767 ^a^	123.793 ^b^	N.D. ^c^	0.009	C32H62O	478.47

RI, retention index; RT, retention time; N.D., not detected; FDR, false-discovery rate; MW, molecular weight. Values represents the means of three replicate, and values within the same row with different superscript letters are significantly different (*p* < 0.05).

## Data Availability

All data are contained within the article.

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
