# Peer review of "Metabolite Variation between Nematode and Bacterial Seed Galls in Comparison to Healthy Seeds of Ryegrass Using Direct Immersion Solid-Phase Microextraction (DI-SPME) Coupled with GC-MS"

_molecules, 2023, doi:10.3390/molecules28020828_

Round 1

Reviewer 1 Report

Metabolite variation between nematode and bacterial seed galls in comparison to healthy seeds of ryegrass using Direct Immersion Solid-Phase Microextraction (DI-SPME) coupled with GC-MS

This work applied DI-SPME coupled with GC-MS to gain information on metabolic profiles associated with nematode and bacterial galls in annual ryegrass to better understand the biochemical interactions between the nematode, bacterium, and plant. This research aims to provide information for the development of detection assays and improvemanagement of disease.

Include in supplementary material:

1.- chromatograms of each of the groups

2.- Mass spectra of at least 2 components.

3.- Calculations of the kobats index.

4.- Include database obtained by GC-MS or mention where these data are stored.

Result and Discussion

Include the supervised analysis, PLS-DA, even try the oPLS for 2 groups between nematode vs healty, healty vs bacterial. It would be interesting to know the differential compounds between the groups.

Materials and Methods.

Include information on DI-SPME, a three-phase fiber (50/30µm DVB/CAR/PDMS, Stableflex 2 cm) for use, such as exposure time, suitable desorption temperature.

Conclusions

They are incomplete, since the requested information is missing.

Author Response

Dear Reviewer,

I would like to thank you from the bottom of my heart for the time and patience which you have given to review our manuscript. I appreciate your highly perceptive comments, which significantly guided us to improve the quality of our manuscript. In the following page, our responses are addressed point to point. Hopefully, we have accepted your suggestions and incorporated all comments and advice.

Thank you

Kind regards

Include in supplementary material:

Point 1.- chromatograms of each of the groups.

Response 1. Chromatograms of each group are added.

Point 2.- Mass spectra of at least 2 components.

Response 2. Mass spectra of two compounds are added as per the suggestion.

Point 3.- Calculations of the kobats index.

Response 3. An Excel file of calculations is added as per the suggestions.

Point 4.- Include database obtained by GC-MS or mention where these data are stored.

Response 4. This database is contained in commercial purchased software and this information is cited in the paper “The data processed with Agilent Mass Hunter Qualitative Analysis software (Version B.08.00, Build 8.0.8208.0) [48]”

 Result and Discussion

Point 5. Include the supervised analysis, PLS-DA, even try the oPLS for 2 groups between nematode vs healty, healty vs bacterial. It would be interesting to know the differential compounds between the groups.

Response 5. PLS-DA and oPLS-DA were plotted between nematode vs healthy and bacteria vs healthy. The chart and description are added into the text as per the suggestions.

 Materials and Methods.

Point 6. Include information on DI-SPME, a three-phase fiber (50/30µm DVB/CAR/PDMS, Stableflex 2 cm) for use, such as exposure time, suitable desorption temperature.

Response 6. The exposure time was 60 min at room temperature and for desorption, the inlet temperature was 290°C and the desorption time was 6 minutes. The same is added to the text as well.

 Conclusions

Point 7. They are incomplete, since the requested information is missing.

Response 7. All the information is tried to include as per your suggestions and sentence are rephrased in the conclusion section.

Reviewer 2 Report

Reviewer report:

This is an interesting and well-written manuscript that falls within the general scope of the journal. The work is - in general terms - well presented, in good scientific English and the experimental work is easy-to-follow.

I suggest acceptance after moderate revision based on the specific comments found below:

Abstract: To my opinion the abstract is rather too general and the authors could add some quantitative analytical data.

Introduction: Although the introductory section is rather "short", it contains all necessary information, including discussion on previously reported methods/techniques for this type of analysis. The authors also mention the advantages of DI-SPME; what about its potential disadvantages compared to headspace SPME?

Results and Discussion: R&D section is structured as a single paragraph, without any sub-sections; is it possible to add a few subsections in order to improve the readability of the manuscript?

Experimental: The authors state that the GC conditions were adopted from [47]. However I was not able to see any discussion on the DI-SPME conditions. If developed in this work the authors should discuss their experiments in the R&D section; if adopted from the literature, this should be clearly mentioned by the authors.

Conclusions: "In this experiment...." please rephrase. 

Author Response

Dear Reviewer,

I would like to thank you from the bottom of my heart for the time and patience which you have given to review our manuscript. I appreciate your highly perceptive comments, which significantly guided us to improve the quality of our manuscript. In the following page, our responses are addressed point to point. Hopefully, we have accepted your suggestions and incorporated all comments and advice.

Thank you

Kind regards

Point 1. Abstract: To my opinion the abstract is rather too general, and the authors could add some quantitative analytical data.

Response 1. Added as per the suggestions.

Point 2. Introduction: Although the introductory section is rather "short", it contains all necessary information, including discussion on previously reported methods/techniques for this type of analysis. The authors also mention the advantages of DI-SPME; what about its potential disadvantages compared to headspace SPME?

Response 2. The information is added as per the suggestion and information is added to the text: However, one of the drawbacks of DI-SPME is associated with its handling like bending, breaking, and stripping of coating which can decrease its lifespan and can be overcome by using HS-SPME to the certain limit [26].

Point 3. Results and Discussion: R&D section is structured as a single paragraph, without any sub-sections; is it possible to add a few subsections in order to improve the readability of the manuscript?

Response 3. Result and discussion section separated and in the result section subheading/ subsections are included as per the suggestion.

Point 4. Experimental: The authors state that the GC conditions were adopted from [47]. However, I was not able to see any discussion on the DI-SPME conditions. If developed in this work the authors should discuss their experiments in the R&D section; if adopted from the literature, this should be clearly mentioned by the authors.

Response 4 The sentence is changed as per the suggestion that it is clearly mentioned that GC conditions were adopted and the DI SPME conditions were addressed in the discussion section and material and method section.

Point 5. Conclusions: "In this experiment...." please rephrase. 

Response 5 Modified as per the suggestion.

Round 2

Reviewer 1 Report

All requested changes were made. Thank you for doing them correctly.